# Star Polycation Mediated dsRNA Improves the Efficiency of RNA Interference in *Phytoseiulus persimilis*

**DOI:** 10.3390/nano12213809

**Published:** 2022-10-28

**Authors:** Zhenhui Wang, Mingxia Li, Ziyi Kong, Endong Wang, Bo Zhang, Jiale Lv, Xuenong Xu

**Affiliations:** Lab of Predatory Mites, Institute of Plant Protection, Chinese Academy of Agricultural Sciences (IPPCAAS), No. 2 Yuanmingyuan West Road, Haidian District, Beijing 100193, China

**Keywords:** Phytoseiidae, nanomaterial, soaking, reproduction

## Abstract

RNA interference (RNAi) is one of the most widely used techniques to study gene functions. There is still a lack of RNAi techniques that can be applied in Phytoseiidae conveniently and efficiently. Star Polycation is a new nanomaterial commonly used as a carrier of dsRNA in RNAi. Five genes of *P. persimilis* (*PpATPb*, *PpATPd*, *PpRpL11*, *PpRpS2*, and *Pptra-2*) were selected to verify whether SPc promotes the delivery of dsRNA into *P. persimilis* through soaking. When each of the five genes were interfered using SPc-mediated dsRNA, the total number of success offspring produced per female in six days decreased by ca. 92%, 92%, 91%, 96%, and 64%. When *PpATPb*, *PpATPd*, *PpRpL11*, or *PpRpS2* was interfered, both the fecundity and egg hatching rate decreased. In contrast, when *Pptra-2* was interfered, reduction in the reproductive capability was mainly the result of the decreased egg hatching rate. Correspondingly, when the target gene was interfered, *P. persimilis* expression of *PpRpL11* reduced by 63.95%, while that of the other four genes reduced by at least 80%. Our studies showed that nanomaterials, such as SPc, have the potential to be used in RNA interference of phytoseiid mites.

## 1. Introduction

Phytoseiid mites (Chelicerata: Arachnida: Acari) play important roles in agroecosystems, as highly effective biocontrol agents of many small pests, including spider mites, whiteflies, and thrips, etc. [1,2,3]. Studies on phytoseiids have mainly focused on their biology, and many interesting phenomena were observed and reported. For example, they have a reproductive pattern called “paternal genome elimination”, referring to males that develop from fertilized eggs but become haploid with their paternal genomes lost at early embryonic stages [4,5]. However, the mechanisms behind such a special pattern are still unclear. Molecular biological research on Phytoseiidae is overall retarded, especially research on gene function, regulation mechanism and pathway are limited [6,7,8,9,10]. One major reason is that their highly ossified small bodies became barriers, thus blocking successful applications of many molecular biological techniques.

RNAi is one of the most widely used techniques to study gene function, regulation, and interaction [11,12]. The first step of RNAi is delivering exogenous dsRNA into the target organism. Commonly used methods include microinjection, oral feeding, and soaking. Overall, microinjection is the most commonly used method in dsRNA delivering. However, microinjection on phytoseiid mites generally cause a high level of physical injury and mortality [13,14]. Oral feeding was the only RNAi method that was relatively successfully applied in Phytoseiidae, with an efficiency ranging from 40% to 80% [15,16]. Since starvation is required for the oral delivery of dsRNA in phytoseiids, this method can only be used when expression of the target gene will not be affected by starvation. In addition, there are high risks that dsRNA may degrade due to an unsuitable pH environment or be shared by nucleases that exist in their alimentary canals [17]. Among the three methods, soaking is the easiest to perform and appears to be less harmful. However, poor interference efficiency was observed in phytoseiid, possibly due to the low permeability of its dense body surface [18].

Nanomaterials are natural, incidental, or manufactured materials with sizes ranging from 1 to 100 nm [19]. Some nanomaterials can bind dsRNA and protect it from degradation due to their chemical stability, minimal cytotoxicity, and biocompatibility, therefore they were used as carriers to transfer exogenous dsRNA into cells efficiently and rapidly [20,21,22,23,24,25]. Star polycation (SPc) is a new nanomaterial synthesized by Li et al. The molecular weight is 19,440 (GPC) and the pdi is 1.07. It is a cationic dendrimer that consists of four peripheral amino acid functionalized arms, with outer shells positively charged (+20.9 mV) to bind the negative charge double strand RNA (dsRNA). This combination could decrease the zeta potential to +4.0 mV by binding dsRNA. Meanwhile, the size of SPc is 100.5 nm and increases to 260 nm when bound with dsRNA [24]. Showing improved delivery efficiency of dsRNA in interfering key genes required for life activities by aphids and black cutworms [26,27,28,29], SPc showed its high potential in pest management [30,31,32,33,34]. It might also be expected to enhance RNAi efficiency through soaking in Phytoseiidae.

*Phytoseiiulus persimilis* is probably one of the most well studied species in Phytoseiidae, but still only a small number of its genes have been studied. Our previous study showed that, when dsRNA was mediated with nanocarrier SPc, it might be delivered into the *P. persimilis* body successfully through soaking [35]. In the present study, five more genes were selected to verify the capability of this nanomaterial in enhancing RNAi efficiency through soaking in *P. persimilis*. Reproductive capabilities of interfered females were evaluated and the relative expression of the target genes were measured. The aim of the present study is to establish a nanocarrier SPc-mediated dsRNA delivery system in *P. persimilis* and provide a potential tool in studying gene functions of phytoseiids.

## 2. Methods and Materials

### 2.1. Mite Colony

*Phytoseiulus persimilis* used in this study were obtained from the colony maintained in the Lab of the Predatory Mite, Institute of Plant Protection, Chinese Academy of Agricultural Sciences. *Tetranychus urticae* Koch (Acari: Tetranychidae), reared on two-week-old soybean seedlings, were used as the prey of *P. persimilis*.

For experimental purposes, *P. persimilis* were reared individually using small units, as described by Zhang et al. [36]. The major piece to make a rearing unit arena is a transparent acrylic board (30 × 20 × 3 mm^3^) with a 10 mm diameter hole in the center. Two pieces of rectangular glass were used to seal the hole, each on one side, creating a 10 (dia.) × 3 mm^3^ arena for *P. persimilis*. A piece of bean leaf was used as the floor, placed between one rectangular glass and the central board. The four layers were clipped together on both ends to avoid mites escaping. Individuals were all reared under 25 ± 1 °C, 60 ± 5% RH, and L:D = 16:8.

### 2.2. Acquisition of the Five Genes and the SPc Mediated dsRNA Complexes

#### Gene Selection

Five genes were selected for the present study. Three of the selected genes, *ribosomal protein L11* (*PpRpL11*), *ribosomal protein S2* (*PpRpS2*), and *transformer-2* (*Pptra-2*), were believed to be involved in *P. persimilis* reproduction regulation [15,16]. Two V-ATPase genes (*VATPb*, *VATPd*) were also selected. V-ATPases are important enzymes in arthropods, and are essential for multiple secretory pathways, from the synthesis and modification of biomolecules to the intracellular transport, secretion, and degradation. ATPases consists of variant subunits determined by different genes [37,38]. These genes were often used as RNAi candidate genes because remarkable changes in biological performances of target organisms were often expected after interference [14,23].

Sequences of *VATPb* and *VATPd* were blasted from the transcriptome (submitted to NCBI) of *P. persimilis*, referring to orthologs in *Aphis gossypii* (NCBI Accession XP_027836958.1) and *Drosophila melanogaster* (NCBI Accession NP_001287401.1). These two sequences were termed as *PpVATPb* and *PpVATPd*. Sequences of *PpRpL11*, *PpRpS2* and *Pptra-2* were identical, as reported by Bi et al. [16].

### 2.3. Gene Cloning and cDNA Synthesis

Approximately 2 μg RNA was extracted from *P. persimilis* (ca. 100 individuals of mixed stages), using MicroElute Total RNA Kit (Omega bio-tek, lnc. R6831-01, Norcross, GA, USA), according to the manufacturer’s recommended protocols. Quality control of extracted RNA were evaluated with spectroscopic quantitation using nanodrop 2000 (Thermo Fisher Scientific, Waltham, MA, USA). cDNA was reverse-transcribed using 1 μg RNA following the instructions of UnionScript First-strand cDNA Synthesis Mix for qPCR (Genesand, Beijing, China).

### 2.4. dsRNA Synthesis

All sequences (Appendix A) were cloned using PCR. The reaction mixture (50 μL) for PCR contained: 25 μL 2 × GS Taq PCR Mix (Genesand, Beijing, China), 1 μL cDNA template, 0.5 μmol forward and reverse primers and nuclear-free water. Conditions of PCR reaction were: 95 °C for 3 min, 35 cycles of 94 °C for 25 s, 58 °C for 25 s and 72 °C for 30 s. PCR products were purified and cloned into pTOPO-Blunt Simple Vector following the instructions of Zero Background pTOPO-Blunt Simple Cloning Kit (Aidlab, Beijing, China). The vectors were transformed into DH5α strain of *Escherichia coli* (Vazyme, Nanjing, China) and the positive single colony were selected for Sanger sequencing (Tsingke, Tianjin, China). Five cloned full sequences were aligned with the transcriptome of *P. persimilis* to make sure the accuracy of nucleotide. Primers used in current study were designed in DANMAN 6.0 and synthesized by Sangon bio-teck (Shanghai, China) (Table 1).

The T7 promoter sequence **5′ TAATACGACTCACTATAGGG 3′** was added in the front of the forward and reverse PCR primers to synthesize the subsequent dsRNA (Table 1). Then, 50 μL of the reaction mixture was made, as described in the PCR reaction. Finally, the dsGFP, dsPpATPb, dsPpATPd, dsPpRpL11, dsPpRpS2, and dsPptra-2 fragments were synthesized according to the instruction of T7 RNAi Transcription Kit (TR102) produced by Vazyme.

### 2.5. SPc Synthesis and RNA Interference

In the present study, SPc was synthesized, as described by Li et al. [24], and provided by the Department of Entomology and MOA Key Lab of Pest Monitoring and Green Management, College of Plant Protection, Chinese Agricultural University. For each gene, SPc and dsRNA were 1:1 mixed to make a complex, with 0.1% tween 20 added. Concentration of each complex was 500 ng/μL. Stability of the complexes were checked using 1% agarose gel electrophoresis, with corresponding unbound dsRNA as the control. The complexes were expected to be blocked with no band presented on the gel.

Approximately 600 newly emerged females were used for interference. According to our previous attempts, ca. 15–20 mites can be soaked in 5 μL formulation [35]. Each soaking lasts for 7 min. For each target gene, ca. 100 *P. persimilis* were interfered in total. Another ca. 100 individuals were interfered using GFP as the control. Five groups were designed. In each group, 15–20 mites were conducted in RNAi on all five target genes simultaneously, with GFP as the control.

After soaking, the complex solution was dried using a piece of filter paper. The mites were left at room temperature until they could move normally. Then, they were placed in rearing arenas, and were reared individually (Figure 1). All individuals were used to evaluate biological performances and measure the relative expression of the target gene.

### 2.6. Change in Reproductive Capabilities of P. persimilis as Affected by RNAi

Each interfered female was provided with a newly emerged male as a mate. Pairs that mated for at least 2 h were reared for 6 days for observing the females’ reproductive capability. Daily fecundity and the hatching rate of eggs were recorded. For each individual, the effective fecundity was estimated as the total number eggs produced in 6 days that hatched successfully. Daily fecundity, hatching rate, and total effective fecundity were compared among the five genes and the control, using one-way ANOVA, with multiple comparisons performed using Tukey’s HSD. For each gene interfered, fecundity and hatching rates were also compared across the 6 days using one-way ANOVA. All statistics were performed using IBM-SPSS 22.0, with diagrams made with GraphPad Prism 8.0. Egg morphology were also observed, photographed using the M205 FCA (Leica Microsystems Ltd., Hessian, Germany).

### 2.7. Relative Expression of the Five Genes in P. persimilis When Interfered

After biological performance observation, all individuals were collected to detect relative expression of the target gene. Our preliminary experiments showed that most changes in the *P. persimilis* reproductive capability due to RNAi could have been observed in 6 days. For each treatment, ca. 15 females were mixed as a replicate, and 4 biological replicates were created for expression analysis. Total RNAs were extracted and the first strand was synthesized using the same method, as described above. CT values of each gene were obtained by real time quantitative PCR. Reaction mixture (20 μL) contained: 10 μL 2 × GS AntiQ qPCR SYBR Green Master Mix, 0.4 μL ROX Dye, 1 μL cDNA template, 0.4 μmol forward and reverse primers, and RNase-free water. Reactions were conducted as follows: 95 °C for 1 min, 40 cycles of 95 °C for 20 s, 58 °C for 20 s and 72 °C for 30 s. *β-actin* gene was used as an internal control. Relative expression of five genes were normalized using 2^−ΔΔCT^ according to the method [39]. For each biological replicate, its expression of the target gene was estimated as the mean of 4 technique replicates. For each gene, relative expression when interfered were compared with the GFP control using the *t*-test (IBM-SPSS 22.0).

## 3. Results

### 3.1. Acquisition of the Five Genes and the SPc Mediated dsRNA Complexes

Sequences of *PpVATPb* and *PpVATPd* were obtained from the transcriptome of *P. persimilis*. The length of these two sequences was 585 and 525 bp, respectively, and encoded 194 and 174 amino acids. All five target sequences were successfully cloned and were identical as blasted sequences in transcripts.

The lengths of all five dsRNAs ranged from 400 to 500 bp. When each dsRNA alone was added into the gel, there was a clear band that showed the same length with the fragments. When combined with the SPc, the SPc mediated dsRNA complexes were blocked in the gel with no band observed (Figure 2). The results were consistent with those reported by Li et al. [24], indicating that SPc can bind dsRNA successfully and stably.

### 3.2. Change in Reproductive Capabilities of P. persimilis as Affected by RNAi

Decrease in reproductive capability were observed in all treatments. When each of the four genes interfered, daily fecundity ranged between 1.9 and 26.3, while the hatching rate ranged between 0.43 to 0.99 (Figure 3A,B). Overall, total effective fecundity of *P. persimilis* in six days decreased by 92%, 92%, 91%, 96%, and 64% when *PpATPb*, *PpATPd*, *PpRpL11*, *PpRpS2*, and *Pptra-2* were interfered, respectively (Figure 3C).

The greatest reduction in reproductive capability was observed when *PpRpS2* was interfered, with 10.2% of individuals becoming completely sterile, and almost all the others stopped laying eggs on the second day. When *PpATPb*, *PpATPd*, and *PpRpL11* were interfered, 44%, 75%, and 70% of individuals stopped laying eggs on the second day, while the proportion of females who stopped laying eggs increased continuously (Table 2). Almost no egg laid after the third day hatched (Table 2). In contrast, when *Pptra-2* was interfered, only ca. 5.3% females stopped laying eggs on the second day. Daily fecundity was only 30% lower than the control, but the egg hatching rate decreased continuously since the third day (Table 3).

### 3.3. Relative Expression of the Five Genes in P. persimilis When Interfered

Relative expression of all five target genes decreased significantly when interfered (Figure 4A–E). Among which, relative expression of *PpRpL11* reduced by 64% (Figure 4C), while that of the other four genes decreased by at least 80%. The maximum decrease in expression (97%) was observed when *Pptra-2* was interfered (Figure 4E).

## 4. Discussion

After being soaked in the SPc mediated dsRNA complex, both the reproductive capability of *P. persimilis* and relative expression of the target genes decreased. These results suggested that SPc mediated dsRNA entered *P. persimilis* body successfully through soaking, which became a RNAi technique that was very convenient to perform. In the present study, expression of *PpRpL11*, *PpRpS2* and *Pptra-2* when interfered were 6%, 70%, and 55% lower, respectively, than those reported by Bi et al. When interferences were performed through oral delivery, suggesting the soaking method is also highly effective [16]. In addition, relative expression of the target genes was measured six days after interference, suggesting the long-term effect of this method.

Previous studies showed that *PpRpL11*, *PpRpS2* and *Pptra-2* are involved in reproduction regulation in phytoseiids [15,16]. Generally, *VATPases* were not considered as directly related to reproduction. However, *PpATPb* and *PpATPd* were expected to be tightly involved in energy metabolism. Arthropod females often had their energetic requirement increase significantly after mating for reproductive purposes [40,41,42]. For example, in wolf spider, the content of Glucose in fertilized female is twice that in virgin individuals [42]. Consumption rates of *P. persimilis* increased by ca. seven times after mating, suggesting there is also high energetic requirement for egg production [43]. When *ATP* genes were interfered, many small arthropods had fecundity reduced due to disturbed energy metabolic [44,45]. It is reasonable to expect similar patterns in *P. persimilis.*

Regulation of reproduction is undoubtedly very complicated. For example, in the model organism *D*. *melanogaster*, there were 30,075 distal regulatory elements related to embryonic development by analyzing cis-regulatory dynamics after egg laying. Gene-expression at different timings during embryonic development also differ [46,47]. In the present study, eggs that failed to hatch show different morphological changes when different genes were interfered. According to the terms generally used in insect embryo development [48], the opacite and transparent parts of each egg were termed as yolk and developmental basis, respectively. When *PpATPb* and *PpATPd* were interfered, almost all eggs failed to hatch had the developmental basis shriveled. When *PpRpL11* and *PpRpS2* were interfered, partial eggs had the yolk shriveled, and some eggs just failed to hatch with no obvious morphological change observed. When *Pptra-2* was interfered, all eggs that produced from the third to sixth day failed to hatch had the yolk shriveled (Figure 5). However, due to limitations of our knowledge, we are currently not able to link the phenotypes to gene functions theoretically.

Studies on molecular mechanism of *P. persimilis* mainly focused on molecular characteristic, protein structure and expression pattern. Limited molecular biological techniques have been applied to phytoseiids successfully in investigating and verifying gene functions. Our study showed that the nanocarrier SPc-mediated dsRNA delivery system can be conveniently and effectively applied to interfere functional genes in phytoseiids, which is the required first step to study gene functions. With this system developed, we expect more techniques, such as fluorescence in situ hybridization, protein–protein interaction, CRISPR, etc., can and will be applied to these tiny organisms, to explain the mechanisms behind their specific biological features.

## Figures and Tables

**Figure 1 nanomaterials-12-03809-f001:**
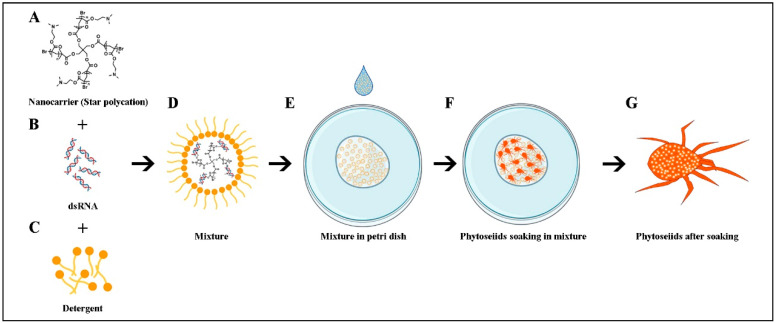
Process of nanocarrier SPc mediated dsRNA delivering through soaking. (**A**) Nanocarrier SPc; (**B**) dsRNA; (**C**) detergent tween 20; (**D**) formulation of SPc, dsRNA and detergent; and (**E–****G**) soaking process.

**Figure 2 nanomaterials-12-03809-f002:**
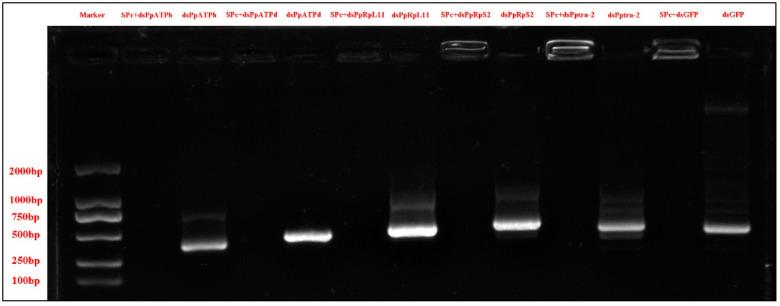
Gel electrophoresis results of SPc mediated dsRNA complexes and corresponding unbound dsRNAs. For each gene, when dsRNA was added, a clear band was observed on the gel. In contrast, no band was observed with dsRNA was mixed with SPc, suggesting that the complex was blocked.

**Figure 3 nanomaterials-12-03809-f003:**
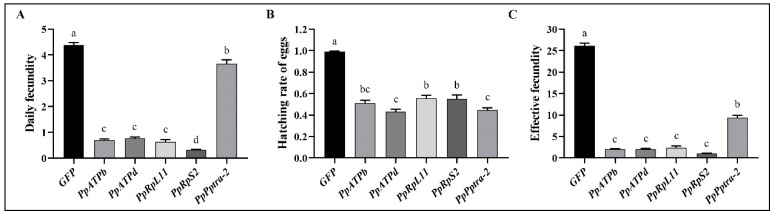
Reproductive capabilities of *P. persimilis* when the five genes were interfered. (**A**). Daily fecundity per female. (**B**) Egg hatching rate. (**C**) Effective fecundity per female. Letters above each bar (Means ± SEM) indicate significant differences across each treatment (SNK tests: *p* < 0.05).

**Figure 4 nanomaterials-12-03809-f004:**
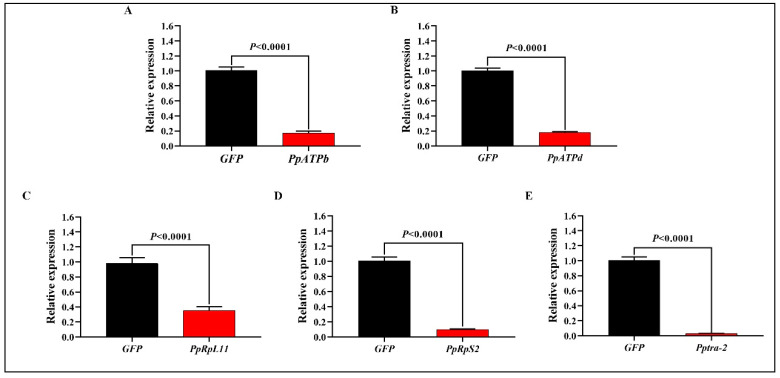
Relative expression of five genes (**A**–**E**) in *Phytoseiulus persimilis* when interfered or not. Total RNAs were extracted from 10–15 *P. persimilis* individuals. Four biological replicates were set for each gene. Differences of relative expression between target gene and GFP were analyzed with a *t*-test using IBM-SPSS 22.0 and all graphs were created using GraphPad Prism 8.0.

**Figure 5 nanomaterials-12-03809-f005:**
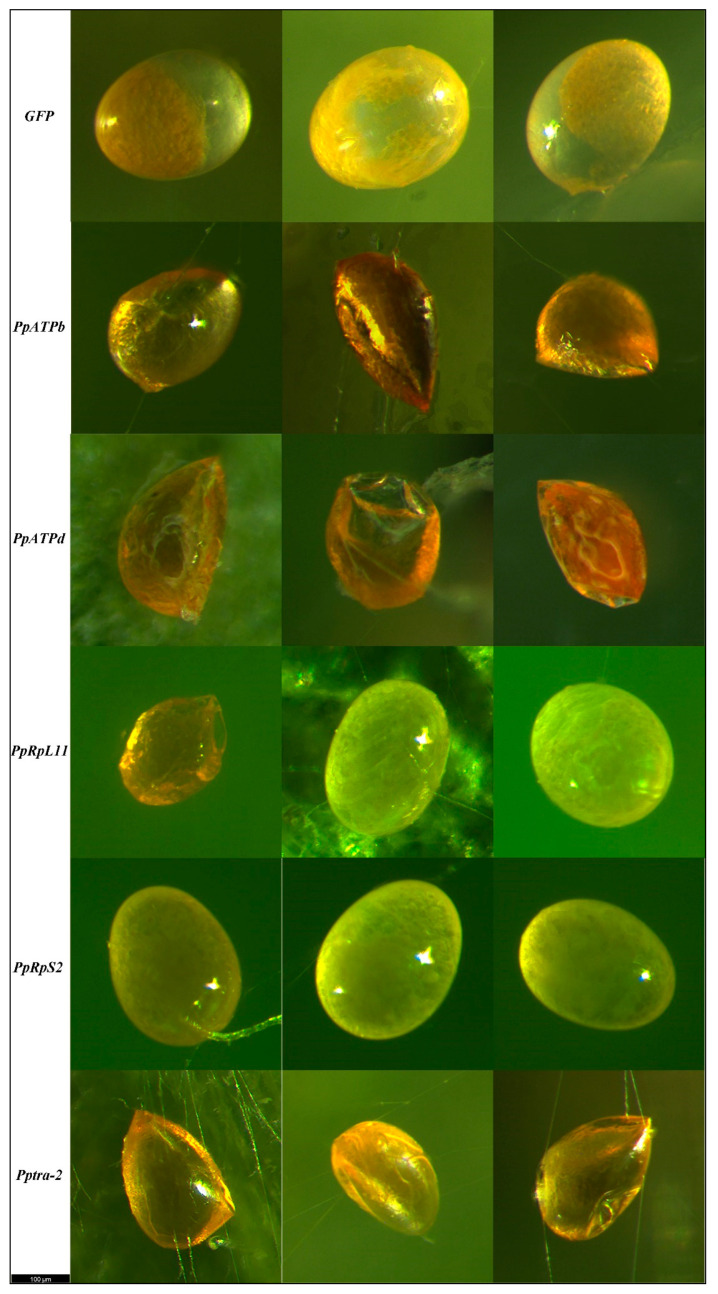
Morphology of normal *P. persimilis* eggs (laid by females interfered with GFP) and unhatched eggs laid by females had each of the five genes interfered. When the mother had *PpATPb* and *PpATPd* interfered, all eggs failed to hatch had the developmental basis shriveled. When the mother had *PpRpL11* and *PpRpS2* interfered, partial eggs had the yolk shriveled, and some eggs just failed to hatch with no obvious morphological change observed. When the mother had *Pptra-2* interfered, all eggs produced from the fifth day failed to hatch had the yolk shriveled. Eggs were photographed using the M205 FCA (Leica Microsystems Ltd., Hessian, Germany).

**Table 1 nanomaterials-12-03809-t001:** All primers used for dsRNA synthesis and RT-qPCR.

Primers	Forward (5′–3′)	Reverse (5′–3′)
dsRNA synthesis		
dsGFP	TAATACGACTCACTATAGGG TGAGCAAGGGCGAGGAG	TAATACGACTCACTATAGGG CGGCGGTCACGAACTCCAG
dsPpATPb	TAATACGACTCACTATAGGG CCCACTCACTGTAGCCAAT	TAATACGACTCACTATAGGG TGTCGTTTACGGAACTCGG
dsPpATPd	TAATACGACTCACTATAGGG ACTGGGTGAAGTTGGCTGA	TAATACGACTCACTATAGGG ATTGCTGAGTCTCGTGGTC
dsPpRpL11	TAATACGACTCACTATAGGG CCGGCAGAGTTCAGAAAGAC	TAATACGACTCACTATAGGG CTACGGTGAGGCACGTTGTA
dsPpRpS2	TAATACGACTCACTATAGGG GACGCTTTTCTTGGAACGAC	TAATACGACTCACTATAGGG CCACAAGTCCGGAGTCAGAT
dsPptra-2	TAATACGACTCACTATAGGG GGAGACGAAGGAAAACGTCA	TAATACGACTCACTATAGGG CGAGTATATCTCCGGCTTCG
RT-qPCR		
*PpATPb*	GAGGATGGGCTTCATACCT	ACGGCAACTCCTGAGAAGA
*PpATPd*	GGTTCGGAAAGAGGAAATG	TCGGCAAGTTTGGGATTC
*PpRpl11*	CGGGAATACGAACTACGC	TCTGCTGGAACCATTTGAT
*PpRpS20*	CAAGGAAGGCGAGAAGG	TGACACCGAGACCAACG
*Pptra-2*	AGATCGGCGTAGCAGGAGT	TCTGGGCATCGTAGACAACC
*actin*	TGGTCGGTATGGGTCAGA	TGGCAGGAGTGTTGAAGGTC

**Table 2 nanomaterials-12-03809-t002:** Daily fecundity of *Phytoseiulus persimilis* when five genes were interfered.

Gene	Daily Fecundity (Mean ± SEM)					
	Day 1	Day 2	Day 3	Day 4	Day 5	Day 6
*GFP*	4.03 ± 0.16 a (n = 79)	5.04 ± 0.13 a (n = 79)	4.98 ± 0.17 a (n = 79)	4.36 ± 0.15 a (n = 77)	3.92 ± 0.15 a (n = 77)	3.89 ± 0.24 a (n = 77)
*PpATPb*	3.32 ± 0.16 a (n = 79)	0.79 ± 0.11 b (n = 77)	0.05 ± 0.05 b (n = 74)	0 b (n = 74)	0 b (n = 74)	0.10 ± 0.07 b (n = 71)
*PpATPd*	3.49 ± 0.14 a (n = 73)	0.48 ± 0.10 b (n = 73)	0.38 ± 0.08 b (n = 73)	0.11 ± 0.06 b (n = 73)	0.06 ± 0.03 b (n = 73)	0.11 ± 0.04 b (n = 72)
*PpRpL11*	2.25 ± 0.11 a (n = 92)	0.61 ± 0.14 b (n = 92)	0.26 ± 0.09 bc (n = 92)	0.21 ± 0.10 bc (n = 91)	0.16 ± 0.09 bc (n = 92)	0.29 ± 0.13 c (n = 91)
*PpRpS2*	1.89 ± 0.10 a (n = 88)	0.02 ± 0.02 b (n = 88)	0.01± 0.01 b (n = 88)	0 b (n = 88)	0 b (n = 88)	0 b (n = 88)
*Pptra-2*	3.60 ± 0.18 a (n = 79)	3.52 ± 0.15 a (n = 79)	2.98 ± 0.20 a (n = 79)	4.32 ± 0.27 a (n = 79)	3.51 ± 0.22 a (n = 79)	4.01 ± 0.27 a (n = 77)

Means within a row followed by different letters are significant differences across each day (SNK tests: *p* < 0.05).

**Table 3 nanomaterials-12-03809-t003:** Daily hatching rate of eggs produced by *Phytoseiulus persimilis* when five genes were interfered.

Gene	Hatching Rate (Mean ± SEM)					
	Day 1	Day 2	Day 3	Day 4	Day 5	Day 6
*GFP*	0.99 ± 0.01 a (n = 77)	0.99 ± 0.01 a (n = 79)	0.99 ± 0.01 a (n = 78)	1.00 ± 0.00 a (n = 74)	0.98 ± 0.01 a (n = 74)	0.99 ± 0.003 a (n = 72)
*PpATPb*	0.62 ± 0.03 a (n = 74)	0.10 ± 0.04 b (n = 41)	0 b (n = 1)	0 b (n = 5)	-	0 b (n = 1)
*PpATPd*	0.56 ± 0.03 a (n = 71)	0.04 ± 0.02 b (n = 46)	0 b (n = 16)	-	0 b (n = 3)	0 b (n = 1)
*PpRpL11*	0.67 ± 0.03 a (n = 83)	0.18 ± 0.07 b (n = 28)	0.08 ± 0.13 b (n = 10)	0 b (n = 5)	0 b (n = 4)	0 b (n = 5)
*PpRpS2*	0.56 ± 0.04 a (n = 79)	0 b (n = 1)	0 b (n = 1)	-	-	-
*Pptra-2*	0.96 ± 0.02 a (n = 69)	0.87 ± 0.03 a (n = 71)	0.40 ± 0.05 b (n = 71)	0.19 ± 0.0 c (n = 63)	0.03 ± 0.02 d (n = 63)	0.07 ± 0.03 d (n = 60)

Means within a row followed by different letters are differences across each day (SNK tests: *p* < 0.05).

## Data Availability

The data used in this study are available on request from the corresponding authors.

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
