# Peer review of "Star Polycation Mediated dsRNA Improves the Efficiency of RNA Interference in Phytoseiulus persimilis"

_nanomaterials, 2022, doi:10.3390/nano12213809_

Round 1
Reviewer 1 Report
The research is mainly showing the utility of nanomaterial SPc in delivering the dsRNA for RNAi assays in Phytoseiulus persimilis, a predaceous mite. The feasibility actually have demonstrated in their previous paper published in 2022. This research used 5 more genes and showed the robustness of the approach. The major problems are the manuscript organization and English writing. The following are my suggestion for improvement.
1. L3-4, In English, the given (first) name comes first, and the family (last) name comes after the given name. The language of this journal is English, not Chinese PinYin. Thus, the English grammar needs to be followed. The same problem found in their previous paper.
2. L58, Li et al (2019) needs to be cited following the citation style (numerically).
3. L72-80 should be moved to Materials and Methods part. Otherwise, describe briefly why they are used. Instead, the objectives should be clearly stated in this part of the Introduction.
4. L92, mm should be separated from the number by a spacer.
5. L98-148 should be separated into 4 parts with subtitles, i.e., one paragraph with a subtitle.
6. L104, the amount (µg) not the volume (µL) of RNA should be used.
7. L132, both this manuscript and their previous paper used the SPc, but neither clearly stated where they obtained the nanomaterial or how they prepared the nanomaterial.
8. L137-148, this paragraph is not clear and needs rephrasing.
9. L156, the ing form of “observe” should be used, i.e., “observing”
10. L167-183, this paragraph should be moved to L153.
11. L190, for sequence alignment, “identical” not “same’ be used.
12. L191-194, more information need to be provided for better understanding Fig 2.
13. L198-213, the citation of the figures in text should be in the same order, i.e., Figure 3A appears first, then Figure 3B, and finally Figure 3C. Otherwise, revise the figure and legends to match the text.
14. L220-224, the letters indicating the statistically significant difference should be superscribed. What does ** indicate for?
15. L226-237, this paragraph should be moved to L197 because it is logical to describe the RNAi effect on the target gene expression and then the effect of the target gene down regulation of the mite development.
16. L275, the Figure 5 is only used for discussion. It should be included in the result.
17. For Pptra2, Figure 3 and Figure 5 seemingly do not match.
Author Response
Dear Reviewer,
Thanks for reviewing my manuscript and sending me the reviewers’ comments. We really appreciate detailed comments provided by the two reviewers, and we have revised the manuscript (marked in red) based on these comments. Please see answers to each comment following in this letter and the revisions in the revised version of the manuscript.
We deeply appreciate your consideration of our manuscript, and we look forward to receiving your decision. If you have any queries, please don’t hesitate to contact me.
Sincerely,
Zhenhui Wang
- L3-4, In English, the given (first) name comes first, and the family (last) name comes after the given name. The language of this journal is English, not Chinese PinYin. Thus, the English grammar needs to be followed. The same problem found in their previous paper.
Changed as suggested. L3-4.
- L58, Li et al (2019) needs to be cited following the citation style (numerically).
Changed as suggested. L63.
- L72-80 should be moved to Materials and Methods part. Otherwise, describe briefly why they are used. Instead, the objectives should be clearly stated in this part of the Introduction.
Changed as suggested. L74-76; L93-102
- L92, mm should be separated from the number by a spacer.
Changed as suggested. L85-87
- L98-148 should be separated into 4 parts with subtitles, i.e., one paragraph with a subtitle.
Changed as suggested. L92; 108; 116; 138
- L104, the amount (µg) not the volume (µL) of RNA should be used.
Changed as suggested. L108
- L132, both this manuscript and their previous paper used the SPc, but neither clearly stated where they obtained the nanomaterial or how they prepared the nanomaterial.
Changed as suggested. L 57-63
- L137-148, this paragraph is not clear and needs rephrasing.
Changed as suggested. L147-157
- L156, the ing form of “observe” should be used, i.e., “observing”
Changed as suggested.
- L167-183, this paragraph should be moved to L153.
In our study, RT-qPCR were performed after reproductive performances of P. persimilis were observed. So, we described relative expression after biological performances based on the experiment procedure. We think it is reasonable, and please let us know if this explanation still does not make sense.
- L190, for sequence alignment, “identical” not “same’ be used.
Changed as suggested. L107, 199.
- L191-194, more information needs to be provided for better understanding Fig 2.
Changed as suggested. L207-209.
- L198-213, the citation of the figures in text should be in the same order, i.e., Figure 3A appears first, then Figure 3B, and finally Figure 3C. Otherwise, revise the figure and legends to match the text.
Changed as suggested. L212-214; L227-231.
- L220-224, the letters indicating the statistically significant difference should be superscribed. What does ** indicate for?
That was typos and we now have deleted “**”.
- L226-237, this paragraph should be moved to L197 because it is logical to describe the RNAi effect on the target gene expression and then the effect of the target gene down regulation of the mite development.
The reasons were stated as comment 10. Effective interferences were expected due to preliminary experiments. In this study, we want to use the same batch of mites for both biological performance and relative expression. For this purpose, we have to do the biological experiments first.
- L275, the Figure 5 is only used for discussion. It should be included in the result.
The major purpose of this study is to verify this method. We briefly observed morphological features of unhatched eggs, but mechanisms were not further explored. More studies are required to provide better explanation. Therefore, we think it is appropriate to put this part in discussion.
- For Pptra2, Figure 3 and Figure 5 seemingly do not match.
In Figure 5, except for the control, only pictures of unhatched eggs were provided. We revised in the manuscript to clarify this. L279-293.

Reviewer 2 Report
Comments on nanomaterials-1968390
In this manuscript, the authors used star polycation to deliver RNAi into P. persimilis through soaking. Overall, the studies were designed well and the results demonstrated the capability of the star polycation for effective RNAi delivery. However, the authors need to address several issues.
1. The authors need to disclose more information about the polymer. If the polymer was synthesized and/or characterized by the authors, related descriptions about the synthesis and/or characterization need to be added in the experimental section. The molecular weight of the polymer should be provided as well.
2. The process of nanoparticle fabrication should be described in the experiment section. The formed nanoparticle needs to be characterized in terms of the hydrodynamic diameter and zeta-potential.
3. The molecular weights or lengths of the dsRNAs and markers in the gel electrophoresis assay need to be added.
4. Scale bars in Figure 5 should be added.
Author Response
Dear Reviewer,
Thanks for reviewing my manuscript and sending me the reviewers’ comments. We really appreciate detailed comments provided by the two reviewers, and we have revised the manuscript (marked in red) based on these comments. Please see answers to each comment following in this letter and the revisions in the revised version of the manuscript.
We deeply appreciate your consideration of our manuscript, and we look forward to receiving your decision. If you have any queries, please don’t hesitate to contact me.
Sincerely,
Zhenhui Wang
- The authors need to disclose more information about the polymer. If the polymer was synthesized and/or characterized by the authors, related descriptions about the synthesis and/or characterization need to be added in the experimental section. The molecular weight of the polymer should be provided as well.
The polymer was synthesized by Li et al (2019). In our studies, this nanocarrier was provided by Department of Entomology and MOA Key Lab of Pest Monitoring and Green Management, College of Plant Protection, Chinese Agricultural University. Information of Star polycation were added in Introduction (Line 57-63).
- The process of nanoparticle fabrication should be described in the experiment section. The formed nanoparticle needs to be characterized in terms of the hydrodynamic diameter and zeta-potential.
Since the present study mainly focused on application of SPc, we don’t think it is necessary to describe its fabrication. Features of the formed nanoparticle has been added in introduction (line 57-63).
- The molecular weights or lengths of the dsRNAs and markers in the gel electrophoresis assay need to be added.
Added in line 206 as suggested.
- Scale bars in Figure 5 should be added.
Changed as suggested. Line 287.
